# AIRE in Male Fertility: A New Hypothesis

**DOI:** 10.3390/cells11193168

**Published:** 2022-10-09

**Authors:** Jana Petrusová, Jasper Manning, Dominik Filipp

**Affiliations:** Laboratory of Immunobiology, Institute of Molecular Genetics of the Czech Academy of Sciences, Vídeňská 1083, 142 20 Prague, Czech Republic

**Keywords:** *Aire* 1, autoimmunity 2, sterility 3, testis 4, Sertoli cells 5, spermatogenesis 6

## Abstract

Male infertility affects approximately 14% of all European men, of which ~44% are characterized as idiopathic. There is an urgency to identify the factors that affect male fertility. One such factor, Autoimmune Regulator (AIRE), a protein found in the thymus, has been studied in the context of central tolerance functioning as a nuclear transcription modulator, responsible for the expression of tissue-restricted antigens in specialized thymic cells that prevent autoimmunity. While its expression in the testes remains enigmatic, we recently observed that sterility in mice correlates with the absence of Aire in the testes, regardless of the deficient expression in medullary thymic epithelial cells or cells of the hematopoietic system. By assessing the Aire transcript levels, we discovered that Sertoli cells are the exclusive source of Aire in the testes, where it most likely plays a non-immune role, suggesting an unknown mechanism by which testicular Aire regulates fertility. Here, we discuss these results in the context of previous reports which have suggested that infertility observed in Aire deficient mice is of an autoimmune aetiology. We present an alternative point of view for the role of Aire in testes in respect to fertility altering the perspective of how Aire’s function in the testes is currently perceived.

## 1. Introduction

The autoimmune regulator (*Aire*) is one of the most studied transcription factors and is known for its critical contribution to the establishment of immunological central tolerance, which occurs in the thymus [1]. Even though the expression of *Aire* in testes was recognized around the same time as in the thymus, its physiological function in this extrathymic organ remains enigmatic. Importantly, *Aire* deficient mice are infertile, but whether this phenotype is a consequence of the lack of *Aire* in the thymus, testes, or both is unclear. What is also puzzling is the exact cellular source of *Aire* in testes i.e., if its role is solely transcriptional and how it’s function is delivered. Several groups have presented data that suggests that testicular *Aire* performs a non-immune function, however their conclusions are not convincing. Therefore, we have decided to compare current published data to data recently generated in our lab. From what we have observed, we hope to bring to the forefront novel ideas regarding the role of *Aire* in testes and its relationship to the pool of *Aire* that is expressed in the thymus.

## 2. Materials and Methods

### 2.1. Breeding Experiment

Male mice used: wild type (*Aire^+/+^*), total body knock-out (*Aire^−/−^*), and males with conditionally depleted *Aire* in spermatogonia stem cells (*Vasa^Cre^Aire^fl/fl^*) and primary spermatocytes (*Smc1β^iCre^Aire^fl/fl^*). All males were housed in pairs with one wild–type female (C57Bl6). The breeding pairs were established at the beginning of their fertility period, i.e., six weeks of age. We established five independent breeding pairs (A–E) and monitored them for six-consecutive months. Each month the number of pups born was recorded. All mice were bred and raised at the animal facility of the Institute of Molecular Genetics in Prague. The protocols for their care were approved by the institutional review committee.

### 2.2. Testes Collection and Statistical Analysis

Testes were taken immediately after euthanizing the animal by cervical dislocation. All males were at the age of six weeks, i.e. at the beginning of their reproductive period. The epididymis and any other tissue (seminal vesicle, fat, etc.) was completely removed from the dissected testes and each testicle was weighted separately. The testis weight represents the average value from two testicles of a single male. We evaluated testicular size and weight from 10 males of each genotype. All graphs were created with GraphPad Prism version 5.00 for Windows (GraphPad Software, San Diego, CA, USA), and data was statistically evaluated by two-way ANOVA analysis.

### 2.3. Collection of Cells from Seminiferous Tubule

Testes from six-week old wild type males were processed as described elsewhere to obtain a single-cell suspension [2]. Briefly, decapsulated testes were treated with collagenase and DNase, followed by filtration. For identification of germ cell populations, we took advantage of Hoechst 33342 staining, which enables the discrimination of the cells according to ploidy [3,4]. Cells were FACS-sorted using a BD Influx high speed cell sorter (BD Biosciences, Franklin Lakes, NJ, USA). At least 10,000 cells per each population were dropped directly into lysis buffer (RLT), followed by the immediate isolation of RNA.

### 2.4. Isolation of RNA and qRT–PCR

Total RNA from FACS-sorted cells was immediately isolated using a commercially available kit (RNeasy MikroKit, QIAGEN, Hilden, Germany). Reverse transcription was done using a RevertAid RT Reverse Transcription Kit (Thermo Fisher Scientific, Waltham, MA, USA) and qPCR with SYBR™ Green PCR Master Mix (Applied Biosystems, Waltham, MA, USA). For the qPCR reaction, we used the LightCycler^®^ 480 System (Roche, F. Hoffmann-La Roche AG, Basel, Switzerland) equipped with software for the analysis of gene expression. The level of gene expression was quantified relative to the house-keeping gene, Casc3. The sequences of primers used in reaction were as follows: Thy1: *Fw* 5’ AAGTCGGAACTCTTGGCACC 3’, *Rev* 5’ TCCAGGCGAAGGTTTTGGTT 3’; Scp1: *Fw* 5’ ACCGTTGGACAACGATTGCT 3’, *Rev* 5’ ATCCATTGCAAGTAAAAGCAACA 3’; Catsper 3: *Fw* 5’ CTTCAGTTTGGCCACGGTTG 3’, *Rev* 5’ CCCAGCTACGGCTACCTCTA 3’; Izumo1: *Fw* 5’GGGATGACCGGTGACTCTTG 3’, *Rev* 5’ ATGCTTCACCCAGCTCAGTC 3’; Wt1: *Fw* 5’ GAGAATCCGCAGGATCGCAG 3’, *Rev* 5’ TGAACTGGCCCGAGAAGTG 3’; and Casc3: *Fw* 5’ TTCGAGGTGTGCCTAACCA 3’, *Rev* 5’ GCTTAGCTCGACCACTCTGG 3’. All mentioned primers annealed at a temperature of 60 °C.

### 2.5. Immunofluorescence and Microscopy

Testes from six-week old wild type males were dissected, immediately immersed in Tissue-Tek^®^ O.C.T. Compound (Sakura Finetek USA, Torrance, CA, USA) and frozen on dry ice. Immunofluorescence detection of proteins was done on cryo-sectioned tissue after fixation and permeabilization. Diluted antibodies were applied to these tissue sections and incubated for 2 h at room temperature. The primary antibodies used were as follows: SOX9 antibody (dilution 1:200, mouse monoclonal sc-166505), AIRE antibody (dilution 1:100, goat polyclonal sc-17986), and SCP3 antibody (dilution 1:200, rabbit polyclonal sc-33195). The origin of all antibodies was from Santa Cruz Biotechnology (Dallas, TX, USA). After several washes, these secondary antibodies were applied to the tissue sections: Alexa Fluor 488 (donkey anti-rabbit), Alexa Fluor 555 (donkey anti-mouse) and Alexa Fluor 647 (donkey anti-goat); all diluted 1:1000 and purchased from Thermo Fisher Scientific (Waltham, MA, USA). Images were taken using a Leica TCS SP5 confocal microscope (Wetzlar, Germany).

## 3. Results

### 3.1. Thymic Aire as an Essential Mediator of Central Immune Tolerance

The story of *Aire* started in 1997, when positional cloning identified mutations in the *Aire* gene as the single genetic defect underlying a disease called autoimmune polyendocrinopathy–candidiasis–ectodermal dystrophy (APECED, OMIM: 240300) [5,6]. APECED belongs to a category of rare human monogenic autoimmune diseases where the immune system malfunctions by attacking the body’s own tissues and organs. It is characterized by multiple autoimmune endocrinopathies, chronic mucocutaneous candidiasis, ectodermal dystrophies, as well as a plethora of symptoms including testicular failure [7]. Several *Aire*-deficient mouse strains were shown to recapitulate human APECED disease, including infertility [8,9]. A major finding explaining the role of *Aire* in the prevention of autoimmunity was that its expression was confined to medullary thymic epithelial cells (mTECs) [10]. In this respect, *Aire* functions as a transcription factor which promotes the “promiscuous” expression of a large repertoire of repressed genes, which are otherwise only expressed in the immune periphery as tissue-restricted antigens [11]. *Aire* achieves this “promiscuous” expression of tissue-restricted antigens by decoding epigenetic marks of repressive chromatin via its PHD1 and PHD2 domains and navigates transcriptional machinery to these silenced genes [12,13,14]. Recent studies estimate that mTECs express ~18,000 genes which represents ~85% of the protein encoded genome. Of these, *Aire* regulates ~4000 genes, which mainly encodes tissue-restricted antigens [15,16,17,18]. mTECs present the products of these genes on its surface as self-antigens that are required for the elimination of self-reactive T cells, or their conversion to Foxp3^+^ regulatory T cell lineage, both of which are required for the prevention of autoimmunity. Thus, the absence or malfunction of *AIRE*, such as that observed in APECED patients, leads to the non-expression and presentation of tissue-restricted antigens on mTECs and the non-removal of self-reactive thymocytes, resulting in the disruption of central tolerance and, consequently, the onset of autoimmunity [19].

### 3.2. Extrathymic Aire Expression

Because of its multiorgan manifestation and severity of autoimmunity, it was originally thought that nearly all symptoms observed in patients with APECED exhibited an autoimmune etiology [9]. However, this would only be credible if *Aire* is exclusively expressed in mTECs. By using various biochemical and genetic approaches, several groups convincingly showed that *Aire* is also expressed on the protein level in secondary lymphoid organs, mainly in the lymph nodes and spleen [20,21,22]. Thus, the question arises whether *Aire* in these extrathymic tissues acts in a similar manner to what has been observed in central tolerance. Indeed, initial studies using *Aire-EGFP*-reporter (*Adig)* mice [23] localized *Aire* expression to a unique subset of lymph node hematopoietic cells which were described as extrathymic *Aire*-expressing cells. The authors concluded that these cells represented a specialized, distinct, and nonconventional APC population that exhibited a CD45^+^ MHCII^+^ CD11c^LOW^ EpCAM^+^ phenotype, influencing the induction of peripheral tolerance in an *Aire*-dependent fashion similar to that mediated by mTECs in the thymus [23,24].

Recently, we conducted experiments exploring the potential phenotypic diversity and the specific function of extrathymic *Aire*-expressing cells. We have shown that the expression of *Aire* protein in lymph nodes is confined to a rare subset of innate lymphoid cells (ILC) that display the characteristics of MHCII^+^ILC type 3 cells [25]. Since *Aire* in MHCII^+^ILC3 cells regulates IL-17 immune responses to *Candida albicans* infections [26], it is clear that *Aire* deficiency in these cells is the underpinning cause of chronic mucocutaneous candidiasis in patients with APECED. Hence, the immune function of Aire as a transcription regulator can be attuned in a cell-specific manner and functionally differs from its well-established role in mTECs in the enforcement of central tolerance.

### 3.3. Aire in Testes and Male Sterility

Interestingly, in addition to lymph nodes and spleen, the expression of the *Aire* protein has also been described in the testes [27,28]. Interestingly, since the testes are not classified as secondary lymphoid organs, a non-immune function of *Aire* could potentially be envisioned. It is important to reiterate that *Aire* whole body knock-out (*Aire^−/−^*) mice are infertile and patients with APECED suffer from testicular failure, which draws an interesting parallel. Thus, the obvious question is whether these reproduction-affected phenotypes are a consequence of Aire malfunction in mTECs, secondary lymphoid organs or testes. Interestingly, Kekäläinen et al. found that sterility in *Aire^−/−^* mouse males could be reversed in the absence of T and B cells, i.e., *Aire^−/−^Rag1^−/−^* mice were found to be fully fertile which led the authors to suggest an autoimmune aetiology of infertility in *Aire^−/−^* males [29]. This finding argues that it is either *Aire* expression in mTECs or in the peripheral hematopoietic system which controls the processes leading to autoimmune attack that results in infertility and precludes the involvement of testicular Aire in this process. However, several reports failed to provide support for such a conclusion.

While these studies confirmed that male *Aire^−/−^* mice suffer from infertility, no prominent lymphocyte infiltrates in the gonads were detected [8]. It was shown that autoantibodies against seminal vesicles and the epididymis appeared between weeks 21–25, although a full cadre was documented between weeks 48–49 [30]. The exception was the detection of autoantibodies against the prostate gland in 7–8 weeks old animals and peaks at 21–25 weeks of age, which was accompanied by pronounced prostatitis, which may affect the quality of sperm and hence fertility [30,31,32]. Thus, while it seems that lymphocyte infiltration, autoantibody production, and autoimmunity towards multiple reproductive targets contributes to infertility in male *Aire^−/−^* mice [33], these phenotypes are observed predominantly later in the life of these animals. However, this is in contrast to the fact that 85% of males are sterile from the beginning of their reproductive cycle, and incapable of producing any litters [9]. Moreover, no evidence has been presented demonstrating that autoantibodies in *Aire^−/−^* mice have a pathogenic role in the onset of autoimmunity [34].

Considering this finding, an alternative view is rather apparent: the primary cause of infertility in male *Aire^−/−^* mice is not the result of the autoimmunity caused by the breach of central tolerance. To validate this notion, the prediction that *Aire* deficiency in mTECs or secondary lymphoid organs (lymph nodes and spleen) does not affect male fertility has already been functionally tested using a mouse strain with the floxed *Aire* allele (*Aire^fl/fl^*). This strain was mated with mice harboring a TEC-specific Cre driver, *Foxn1^Cre^*, or with a strain with a hematopoietic-specific Cre driver, *Vav1^Cre^* [35]. A detailed analysis revealed a significant decrease of *Aire* mRNA in the thymus of *Foxn1^Cre+^Aire^fl/fl^* and secondary lymphoid organs of *Vav1^Cre+^Aire^fl/fl^* animals. The expression of *Aire* in testes from both *Foxn1^Cre+^Aire^fl/fl^* and *Vav1^Cre+^Aire^fl/fl^* was comparable to testes from *Aire* sufficient animals. Importantly, both *Foxn1^Cre+^Aire^fl/fl^* and *Vav1^Cre+^Aire^fl/fl^* males were fully fertile. Since Aire protein expression had been localized only to the thymus, secondary lymphoid organs, and testes, the infertility observed in male *Aire^−/−^* mice correlated with the absence of *Aire* expression in the testes and not with its deficiency in the thymic stroma or hematopoietic system [35]. To the best of our knowledge, this experiment provided the first genetic evidence for a functional link between fertility and *Aire* expression in the testes.

### 3.4. Which Cells in Seminiferous Tubule Express Aire?

The paradigm-changing observation described above leads to one question: what is the mechanism(s) by which *Aire* deficiency in the testes initiates an animal’s infertility? The first part of this question concerns the precise cellular source of *Aire* in the testes, which then allows one to consider the possible scenarios regarding how the function of *Aire* is delivered to support fertility.

Thus far, the precise cellular localization of the Aire protein production in the testes remains highly controversial. Notably, Schaller et al. claimed to find Aire protein in the nuclei of germ cells, specifically in early spermatocytes and spermatogonia but not in spermatids or Sertoli cells. For the immunohistochemical detection of Aire, the authors used the Aire specific monoclonal antibody, clone 5H12, which is routinely used by many labs for detection in thymic medullary cells [27]. While the cells which they stated contained Aire were located close to the inner peripheral circumference of the seminiferous tubules, their identification was only positional, with no cell specific colocalization marker (Smc1β for early spermatocytes or Thy1 for spermatogonia stem cells). Therefore, the only conclusion which can be made from this study is that *Aire* expression in the testes is confined to the seminiferous tubules.

Radhakrishnan et al. found that *Aire* mRNA is expressed in all evaluated stages of germ cell development, including the spermatogonia from neonatal and adult testes, primary and secondary spermatocytes, and spermatids [28]. The authors also showed that the *Aire* expression pattern in mouse testes is dynamic, with the highest level of *Aire* mRNA observed during the first week of life [28]. The authors proposed that this pattern of expression correlates with the requirement for Aire abundancy during the first wave of spermatogenesis. Interestingly, four weeks after birth, a dramatic ~10-fold decline in *Aire* expression was observed as the animals sexually matured. However, the presence of *Aire* mRNA and its decrease between the first and fourth week of life did not correlate with protein levels where no changes were seen [28]. Nevertheless, the common denominator in both studies discussed above is the suggestion that *Aire* is expressed in germ cells residing within seminiferous tubules.

If the cellular source of testicular Aire in the testes is indeed in germ cells and this pool of Aire is required for fertility, then the *Aire* deficiency in germ cells should recapitulate the infertility observed in male *Aire^−/−^* mice. To test this prediction, we again utilized *Aire^fl/fl^* mice which were crossed with animals that expressed either *Vasa^Cre^* or *Smc1β^iCre^* drivers. The mouse vasa homologue (*Vasa*) gene showed spermatogonia stem cell expression [36] from embryonic day 12.5 onwards after entering the gonadal anlage [37]. The Smc1β gene product is a member of a cohesion complex that forms a ring structure around meiotic chromosomes and is expressed at the onset of meiosis I of primary spermatocytes (SCI). *Vasa* as well as *Smc1β* are specifically expressed in germ cells but not in testicular somatic cells [38]. Thus, the utility of the *Vasa* and *Smc1β* promoters as Cre drivers makes them suitable for conditional depletion of desired floxed alleles with a specific effect on spermatogonia stem cells and primary spermatocytes, and their progeny-secondary spermatocytes (SCII), round spermatids and sperm (RS+S). *Vasa^Cre^Aire^fl/fl^* and *Smc1β^iCre^Aire^fl/fl^* males were cross-bred with wild type females over a six-month period. The breeding performance was closely monitored and compared to wild-type couples. There was no significant difference in fertility between the two transgenic males, whereas the total body *Aire* KO gave birth to only a few pups at the beginning of the reproduction cycle (Figure 1A). The size and weight of the testes correlated with the results (Figure 1B) with no significant difference observed between wild-type, *Vasa^Cre^Aire^fl/fl^* and *Smc1β^iCre^Aire^fl/fl^* males. These parameters of *Aire^−/−^* male testes were significantly reduced to 55–80% when compared to wild-type males (Figure 1C).

It is important to note that, in order to identify Aire expressing cells in testes, Radhakrishnan and colleagues FACS-sorted cell populations from the flow cytometric profile of spermatocyte populations stained with Hoechst 33342. However, the generated Hoechst profile seemed rather non-standard when compared to previously published analyses [3,4,39]. Additionally, it is also unclear as to which cell types were sorted and analyzed [28]. This caveat weakens the conclusion that Aire mRNA is present in all germ cell types. Moreover, the authors omitted the analysis of Sertoli cells, which are also present in the seminiferous tubules, as a potential source of Aire.

To precisely identify the cellular source of Aire, we isolated all germ cell types present in seminiferous tubules (Figure 2A), along with Sertoli cells [2]. All indicated cell types were FACS-sorted directly into the lysis buffer for isolation of mRNA. Using RT-qPCR analysis for each sorted cell type, we quantified *Aire* transcript levels and utilized the following cell-type specific marker genes: *Thy1* (Thy1 cell surface antigen; [40]) in spermatogonia stem cells, *Scp1* (Synaptonemal complex protein 1; [41]) in primary spermatocytes, *Catsper 3* (Cation Channel Sperm Associated 3; [42]) in secondary spermatocytes, *Izumo1* (Izumo Sperm-Egg Fusion 1; [43]) from round spermatid and sperm, and *Wt1* (Wilms Tumor Protein; [44]) for Sertoli cells. Unexpectedly, we found that Aire was exclusively present in Sertoli cells. All other cell types showed a 10,000–100,000 times lower expression of *Aire* mRNA in comparison to Sertoli cells (Figure 2B). In addition, Aire on the protein level is also detectable only in Sertoli and no other cells in the seminiferous tubule (Figure 2C).

Our finding is further supported by the recent work of Grive et al. who identified the genes specifically expressed by meiocytes in the first and second round of meiotic division as well as the population of spermatogonia stem cells [45]. They not only identified conserved markers of germ cells but also their novel genes. Nevertheless, Aire was not detected in any of the described spermatogenic populations (GEO: GSE121904, blasted Aire exon 1 in dataset) [45,46,47]. In contrast, we identified an in silico *Aire* gene expressed in highly purified Sertoli cells from a transgenic SOX9-GFP male dataset. The Aire transcript was present beginning at day 10, and higher levels were observed from day 18 onwards during male development (GEO: GSE59698) [48].

## 4. Discussion

The main objective of this article was to compare our recently generated data to previously published data sets, some of which present inconsistent or unclear outcomes that describe the role of Aire in male fertility. Our work is an alternative to recent formulated thoughts of the potential role of Aire in testes, which are based on previously published data that Aire in testes is localized to germ cells [50]. Recent genetic data [35], in combination with our presented results, support the view that it is the Aire in Sertoli cells which might be important for the maintenance of fertility. Even if Aire is present in spermatogonia stem cells or primary spermatocytes as suggested by other studies [27,28], its depletion in these cells would not impact male fertility. This alternative point of view regarding the function of testicular Aire provides a new perspective on the role of Aire in Sertoli cells in respect to the control of male fertility.

What could be the possible role and mechanism of Aire in testes? Some insight may be provided by *Foxn1^Cre+^Aire^fl/fl^* mice which appear to be fully fertile for at least the first twelve months of life [2,35]. This provides evidence that even in the absence of Aire in the thymic stroma resulting in the continuous process of multiorgan autoimmunity, the testes are not functionally affected. Thus, since infertility which arises as a result of autoimmunity can only occur in the absence of Aire in the testes and later in life, this may provide a clue for the role of testicular Aire. Notably, the testes are viewed as an immune privileged organ in which Sertoli cells are critical for the generation and maintenance of a barrier within seminiferous tubules [51]. Aire in Sertoli cells could be directly involved in this process either as a transcription factor which regulates the expression of genes required for barrier function or some other function which has yet to be determined. Consistent with this proposal, since the barrier in *Foxn1^Cre+^Aire^fl/fl^* mice remains intact and provides protection from autoimmune attack, it maintains fertility throughout life. However, this barrier in *Aire^−/−^* mice may be improperly formed and non-functional allowing the penetration of autoantibodies and autoreactive cells to be critical reproduction targets. Since in *Aire^−/−^* mice an autoimmune attack is the result of *Aire* deficiency in mTECs, the accompanying lack of testicular *Aire* is not a direct causative factor of infertility but rather a consequence of its specific role in Sertoli cells. While this scenario would explain the onset of infertility with autoimmune aetiology later in life, it still does not provide an explanation which states why the vast majority of *Aire^−/−^* mice cannot reproduce at all. Clearly, other mechanisms dictate how Aire in Sertoli cells modulate fertility, which warrants further investigation.

Another aspect of this issue is the fact that dominant mutations in the Aire gene affect approximately 1 in 550 people [12]. However, the effect of dominant mutations in Aire on patient fertility has not been adequately addressed.

## 5. Conclusions

Together, the presence of Aire in testes challenges the traditional view of Aire as a unique immune-related transcription factor and should be an invitation to members of this field of study to revisit its physiological role in a much broader context examining multiple regulatory mechanisms that operate in a cell/tissue-dependent fashion. In this context, it was our intention to provide a conceptually new hypothesis that could begin to resolve the issue of Aire-mediated fertility from a completely different perspective. As our studies continue to progress, future experimentation and analysis of a generated conditional Aire knock-out in Sertoli cells will be an import step in strengthening our current data identifying Sertoli cells as the source of Aire in seminiferous tubules.

## Figures and Tables

**Figure 1 cells-11-03168-f001:**
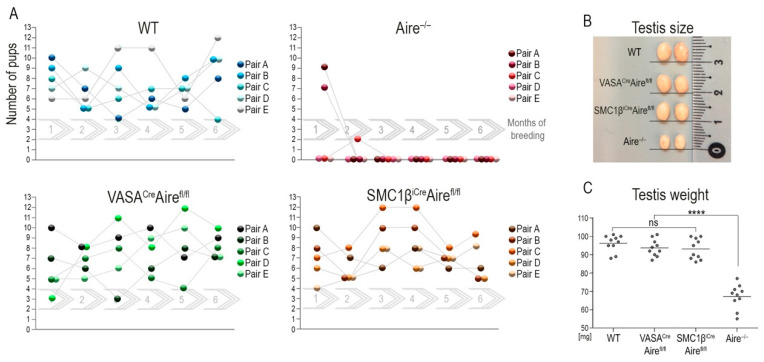
Depletion of Aire in spermatogonia stem cells and spermatocytes does not affect male fertility. (**A**) Male mice with conditionally depleted Aire in spermatogonia stem cells (*Vasa^Cre^Aire^fl/fl^*) and primary spermatocytes (*Smc1β^iCre^Aire^fl/fl^*) exhibit fertility that is comparable to wild-type males in terms of the number of pups per litter. In contrast, whole-body Aire knock-out (*Aire^−/−^*) mice produced only a few pups at the beginning of their reproduction period. We established five independent breeding pairs (A–E) and monitored them for six-consecutive months (1–6, gray continuous arrows). (**B**) The size of the testes in *Vasa^Cre^Aire^fl/fl^* as well as *Smc1β^iCre^Aire^fl/fl^* males was fully comparable to WT controls. In contrast, six–week–old *Aire^−/−^* males had significantly smaller testes. (**C**) The decrease in testes size was confirmed by assessing their weight. *p* < 0.0001 = **** and ns = not significant.

**Figure 2 cells-11-03168-f002:**
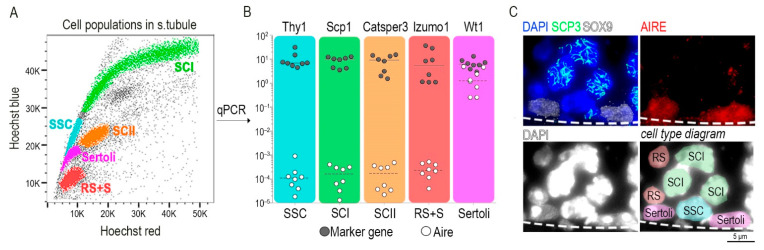
Sertoli cells express Aire-coding mRNA. (**A**) The cell population from within the seminiferous tubule was stained with Hoechst 33342 and visualized in Hoechst red and blue channels. A color code was applied to the following cell types: turquoise spermatogonia stem cell (SSC), green primary spermatocytes (SCI), orange secondary spermatocytes (SCII), red round spermatids and sperm (RS+S), magenta Sertoli cells (Sertoli). (**B**) A single cell suspension from (**A**) was FACS-sorted and the indicated cell-types were prepared for mRNA isolation and qRT–PCR. Only Sertoli cells expressed physiologically relevant levels of Aire mRNA. The *y* axis represents the relative transcript levels of *Aire* and marker genes normalized to the *Casc3* gene. (**C**) Protein immunodetection on the testicular tissue-sections showed the localization of AIRE in the nuclei of Sertoli cells (upper right panel) identified by their positivity to SOX9 (white color, upper left panel). SCP3 is a marker of primary spermatocytes (SCI). Spermatogonia stem cells (SSC), typically positioned between two Sertoli cells at the base of seminiferous tubule and round spermatid (RS), are identified as DAPI^+^SOX9^−^SCP3^−^, exhibiting a typical nuclear morphology [49]. The dashed line demarcates the edge of the seminiferous tubule. Cell types within the diagram (lower right panel) correspond to the color scheme used in qPCR.

## Data Availability

Not applicable.

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
