# Peer review of "AIRE in Male Fertility: A New Hypothesis"

_cells, 2022, doi:10.3390/cells11193168_

Round 1

Reviewer 1 Report

See attached PDF file. I have discussed with the handling editor that it is OK to review non-anonymously.

Reviewer 2 Report

The manuscript from Petruskova and colleagues provides an alternative point of view for the role of Aire in testes with respect to fertility altering the perspective of how Aire function testes are currently perceived. The manuscript is very interesting and well-written. I have only a few suggestions.

1-     The authors have not clearly indicated whether infertility is due to a lack of spermatozoa in the semen (that is, in the vas deferens of the mice analyzed) or if specific defects of spermatogenesis have been identified.

2-  Regarding the cellular localization of AIRE protein, double immunofluorescence using a specific marker of Sertoli cells (e.g. INSL3) should be used. Or the immunoelectron microscopy could definitively prove which are the testicular cells expressing this protein.

Reviewer 3 Report

In this manuscript, authors presented well defined and scientifically sound hypothesis with substantial new information as the first genetic evidence for a functional link between fertility and Aire expression in testes. Experimental study provides information regarding presence of Aire in Sertoli cells.

The Title of the manuscript is concise and relevant. The aim and scope of the study explained well. Introduction is quite comprehensive and highlighted work importance. Although, materials and methods needs improvement (should be descriptive). However, authors provided well explained interpretation of results and discussion. Overall, the hypothesis is nicely written.

Before proceeding further, I expect the authors to thoroughly proofread the document and fix all grammatical and typographical errors (some examples include L214, L250, L262).

Minor suggestions:

1) In L124, reference number in square bracket will placed next to the cited author’s name (ex: Kekalainen et al. [29]). Same as in L158

2) In materials and methods, repetition of lines must be avoided (2.1., L47,48 and 2.2., L 54-56), or provide it separately with new heading as ‘statistical analysis’
